# Collaborating to Improve Neonatal Care: ParentAl Participation on the NEonatal Ward—Study Protocol of the neoPARTNER Study

**DOI:** 10.3390/children10091482

**Published:** 2023-08-30

**Authors:** Hannah Hoeben, Milène T. Alferink, Anne A. M. W. van Kempen, Johannes B. van Goudoever, Nicole R. van Veenendaal, Sophie R. D. van der Schoor

**Affiliations:** 1Department of Paediatrics/Neonatology, OLVG, 1091 AC Amsterdam, The Netherlands; h.hoeben@olvg.nl (H.H.); m.t.alferink@olvg.nl (M.T.A.); a.a.m.w.vankempen@olvg.nl (A.A.M.W.v.K.); n.r.vanveenendaal@olvg.nl (N.R.v.V.); 2Department of Paediatrics, Emma Children’s Hospital, Amsterdam UMC, Vrije Universiteit Amsterdam, 1105 AZ Amsterdam, The Netherlands; h.vangoudoever@amsterdamumc.nl; 3Department of Neonatology, Wilhelmina Children’s Hospital, 3508 AB Utrecht, The Netherlands

**Keywords:** family-integrated care, family-centred rounds, neonatology, parental participation, parental stress, shared decision-making, family centred care, patient empowerment, biomarkers of stress

## Abstract

Parents are often appointed a passive role in the care for their hospitalised child. In the family-integrated care (FICare) model, parental involvement in neonatal care is emulated. Parental participation in medical rounds, or family-centred rounds (FCR), forms a key element. A paucity remains of randomised trials assessing the outcomes of FCR (embedded in FICare) in families and neonates, and outcomes on an organisational level are relatively unexplored. Likewise, biological mechanisms through which a potential effect may be exerted are lacking robust evidence. Ten level two Dutch neonatal wards are involved in this stepped-wedge cluster-randomised trial FCR (embedded in FICare) by one common implementation strategy. Parents of infants hospitalised for at least 7 days are eligible for inclusion. The primary outcome is parental stress (PSS:NICU) at discharge. Secondary outcomes include parental, neonatal, healthcare professional and organisational outcomes. Biomarkers of stress will be analysed in parent–infant dyads. With a practical approach and broad outcome set, this study aims to obtain evidence on the possible (mechanistic) effect of FCR (as part of FICare) on parents, infants, healthcare professionals and organisations. The practical approach provides (experiences of) FICare material adjusted to the Dutch setting, available for other hospitals after the study.

## 1. Introduction

Due to the technological environment of the modern neonatal ward, it is a worldwide practice that preterm or ill infants and their parents are commonly separated, and both physical and emotional closeness are impaired [1]. Having an infant hospitalised in a neonatal intensive care unit (NICU) is a stressful experience for parents [2]. Parents are generally assigned a supportive role during the infant’s hospital stay [3]. As a result, many parents feel anxious, stressed and unprepared to care for their infants during hospitalisation and after discharge [1,4,5,6]. Specifically in the case of preterm delivery, the unexpected birth at a lower gestational age causes mothers to have less time for emotional preparation [7]. Parents experience more stress related to feelings of helplessness, exclusion, alienation, anxiety and depression [1,8]. These symptoms tend to persist after discharge and can affect the short- and long-term relationship with their infant [9,10]. Developmental research has firmly established the quality of the relationship between an infant and his or her parents as an important factor influencing the child’s later development. When children develop a secure and supported relationship with their parents or caregivers in the first years of life, they generally have better cognitive outcomes and better social interactions; they display fewer behavioural problems, and achieve better at school [11].

When examining sources of parental stress during admission of a critically ill infant, role alteration is found to be of great importance. The severe illness and the NICU environment inhibit parents from naturally growing into the new role of being or becoming a parent and parent–infant closeness is inhibited [12]. This can make them feel like a visitor instead of caregiver with feelings of helplessness about how to best support their infant during this stressful period [8]. By educating parents in their irreplaceable role and how to participate in neonatal care, healthcare professionals can help parents gain more confidence and reduce their levels of stress [13]. However, this close collaboration with parents is just starting to arise, and still needs to be implemented and integrated into work processes and culture of many NICUs and neonatal wards [12].

The family-integrated care (FICare) concept encourages and educates parents to provide the care for their infant as much as they are able and their confidence allows [14]. The FICare model is a collaborative program designed to support parents to cope with the distress that comes with their newborn’s hospital admission, based on four pillars: educating parents, educating healthcare professionals, peer-to-peer support and the unit environment. By empowering parents, the goal of FICare is to prepare them emotionally, cognitively, and physically to care autonomously for their infant at the time of discharge [15]. Parents are guided in their learning process as primary caregivers for their hospitalised infant by recognizing their valuable contributions and treating them respectfully as essential members of the care team, thus enhancing their sense of competence [16].

Healthcare professionals may experience a role change with the implementation of FICare, from being hands-on caregiver for the infant to becoming a coach for parents. In this collaboration, the care team is expanded with a “new” expert, i.e., the parent, who participates in the shared medical decision-making [17]. Education and support for the healthcare professionals provides them with the tools for coaching and adaptation to their new role.

Communication between healthcare professionals and parents in the neonatal ward has positive and negative effects on parents’ coping, knowledge, participation, parenting and satisfaction [18]. FICare aims to ameliorate parent–provider communication, which can have an effect on parental stress, by increasing the parents’ confidence and reducing their anxiety [18]. Regarding adequate communication, parental participation in medical rounds, also known as family-centred rounds (FCR), is a key element of the FICare concept [16,19].

Family-centred rounds include parents on medical rounds, aiming to involve them in the process of patient management. It allows parents to hear the development in their infants’ medical condition first-hand from the healthcare professionals and allows them to ask questions [20,21]. Even more important, parents can actively participate in the rounds to provide healthcare professionals with additional medical information on the current clinical state of their infant. Parents are excellent observers of their child, and often the most continuous factor in the care of their infant in the neonatal ward. Active participation in medical rounds is a logical next step and gives parents the opportunity to participate in shared decision-making.

The FICare model, including FCR, has been shown to improve neurodevelopmental outcomes in infants admitted to NICUs and level two neonatal wards in non-European settings [22,23,24]. This effect might be mediated by the reduction in parental stress levels caused by FICare during the neonatal period [22,25,26]. The underlying biological mechanisms of this reduced-stress response have not been previously investigated. Evidence is also not yet available on the effects of FICare in a European (Dutch) setting, differing from Canadian or American settings in many (logistic) ways. For example, many (relatively) smaller hospitals exist in the Netherlands and transfers of neonates between those hospitals are very common. Many healthcare professionals find it difficult to adjust the (extensive) FICare concept to their own setting. In order to prevent the emergence of great variations in FICare practice between hospitals, there is a need for a universal programme, adapted to the Dutch situation.

With the present study, we aim to investigate the effect of the implementation of FCR, incorporated into the FICare principles, on parental stress at discharge using a multicentre stepped-wedge cluster-randomised trial among parents of infants admitted to level two neonatal wards in the Netherlands. We hypothesise that the implementation of FCR embedded in FICare is superior to standard neonatal care without FCR with regard to the primary outcome. We will use a universal implementation strategy of FICare, adapted to the Dutch setting, to provide a practical approach for other Dutch hospitals after the study is concluded. Secondary outcomes include outcomes at the individual level in parents (longitudinal assessment of mental wellbeing) and infants (during the first year of life), and outcomes at the cluster level (effects on healthcare professionals, adherence to intervention and cost-effectiveness). Alongside, we aim to gain more insight into the underlying mechanism of the (expected) effect by measuring biomarkers of stress in parents and infants.

## 2. Materials and Methods

### 2.1. Patient Organisation Involvement

The Dutch patient organisation (Care4Neo) is actively involved in all stages of study, starting from the study design to the implementation of FICare in the participating hospitals, project evaluation and dissemination of the results. The Guideline for Reporting Involvement of Patients and the Public (GRIPP2 short form) is followed [27]. In the grant proposal, two representatives from Care4Neo (S.O-B., M.V.) were included in the project group, ensuring that the patient perspective and relevance were duly considered.

Regarding the study protocol and outcomes measured in the trial, a national survey was conducted together with Care4Neo to prioritise outcomes based on parents’ perspectives. The survey was disseminated through social media platforms of Care4Neo. Care4Neo was actively involved in the final selection of measurement methods and questionnaires. S.O.-B. and M.V., together with Care4Neo’s Parent Advisory Board consisting of experienced parents, thoroughly reviewed the questionnaires, including explanatory texts, and provided input on how to inform parents about any abnormal scores on the questionnaires during study participation. Additionally, Care4Neo played a key role in revising the patient information forms. During the trial, Care4Neo is involved in the implementation of FICare by coordinating the peer-to-peer support programme in each participating centre.

### 2.2. Design

Due to the nature of the intervention, which involves changes to unit-level provision of care (medical rounds) and interaction between participants, there is a risk of cross-contamination. Therefore, to avoid contamination of patients and staff, the stepped-wedge cluster-randomised controlled trial design was selected. The trial is designed according to the CONSORT 2010 guidelines for stepped-wedge cluster-randomised controlled trials [28]. Randomisation will be performed using a random number generator, which will randomly assign the timing of start of intervention between sites. Hospitals will be stratified by level of care delivered (post-intensive care or not). The program will generate a series of blocks of varying sizes for each stratum and allocate units a time for intervention (3, 6, 9, 12 or 15 months, respectively) as is visualised in Figure 1.

### 2.3. Setting

In the Netherlands, the level of neonatal care is classified similarly to the structure provided by the American Academy of Pediatrics (AAP) [29]. Level 2 NICUs, also called special care nurseries, care for babies born at a gestational age of 32 weeks or more. These facilities are suited for babies who have moderate medical issues and are expected to recover fairly quickly. All the participating centres are level 2 neonatal wards. The average number of patients admitted to the neonatology department for more than 7 days per participating centre varies from 80 to 300 patients per year.

### 2.4. Study Population

In 2021, in the Netherlands, approximately 15% of (live) born infants were born preterm and/or small for gestational age, accounting for 25,018 infants [30]. According to Dutch policies, infants born with a gestational age < 32 weeks and/or birthweight < 1200 g or in need of intensive care (e.g., cardiorespiratory support) are born in/or treated at a NICU (level 3). In a Dutch level 2 neonatal ward, infants may receive non-invasive respiratory support, have central venous catheters and receive multiple medications. It is quite common for infants to be transferred between different hospitals prior to their final discharge to home.

In this paper, the intervention group will be referred to as the FICare-group and the control group as the standard neonatal care (SNC) group.

#### 2.4.1. Inclusion Criteria

Recruited patients will be patients admitted to a level 2 neonatal ward for 7 days or more. To be eligible to participate in this study, a subject (parent–infant dyad/triad) must meet all of the following criteria:−Infant requiring hospital admission directly (within 24 h) after birth;−Parent are 18 years or older;−Written, informed consent of both parents/legal guardians (compliant to the regulations of the Central Committee on Research Involving Human Subjects).

All healthcare professionals (nurses, nurse practitioners, residents, paediatricians and neonatologists) that are employed on the neonatal ward during the study are eligible to participate regarding healthcare professional outcomes.

#### 2.4.2. Exclusion Criteria

A potential subject (parent–infant dyad) who meets any of the following criteria will be excluded from participation in this study:−Infant’s hospital stay shorter than 7 days;−Infant with severe congenital or syndromal anomaly;−Infant with critical illness who is unlikely to survive;−Parents with current severe psychosocial problems;−Involvement of child protective services in the family;−Parents not able or not willing to fill out questionnaires in English or Dutch.

Healthcare professionals of whom the employment in a participating centre is discontinued through the course of the study, will be excluded from further measurements.

### 2.5. Procedures

All ten participating hospitals start the control phase, delivering SNC as they did prior to the study. Subsequently, randomisation will determine in which order the neonatal wards will start with the implementation of FCR according to the FICare principles. The outcomes will be measured at the individual level (parental and neonatal outcomes) and at the cluster level (healthcare professional-outcome measures, adherence to the intervention and cost-effectiveness). Given that the intervention includes active participation of parents in neonatal care and medical rounds, blinding is not possible for participants and researchers. However, as the outcomes cannot be influenced by the researcher and are unlikely to be influenced by the parents’ knowledge of the intervention, the risk of performance and detection bias is assumed very low.

In the three months preceding the intervention period (the implementation period), FCR and FICare principles will be implemented on the neonatal wards of the participating hospitals. We will develop and use a universal implementation strategy of FICare, adjusted to the Dutch setting, in close collaboration with the Dutch neonatal parent and patient advocacy organisation (Care4Neo). To ensure feasibility and sustainability of the intervention in all participating sites, an FICare-working group will be formed at each site. Each participating centre is responsible for the recruitment of participants of their own FICare-working group. To achieve the broadest possible representation, FICare-working groups will include neonatal nurses, paediatricians, veteran parents and other neonatal healthcare professionals (such as psychologists or physiotherapists). Veteran parent(s) will be recruited through the patient organisation (Care4Neo) or via the outpatient clinic by paediatricians. Training and guidance of the veteran parents will be coordinated by Care4Neo. The FICare-working group will be responsible for tailoring the intervention to the specific neonatal ward, in cooperation with the research team, and the planning and execution of the staff and parent training.

The four pillars of FICare will be translated into a package of education and other materials adapted to the Dutch setting, to support FICare in all participating sites, including the following components [15]:Education of parents:−Information on the hospital admission and care of an infant, including a digital application (NeoZorg application);−The possibility to keep track of (medical) information in a diary and/or digital application (NeoZorg application, see further);−Educational and thematic meetings (physical and/or digital) with other parents, led by either healthcare professionals (nurses, paediatricians and/or paramedic staff) or veteran parents.Education of healthcare professionals:−Comprehensive e-learning on FICare and FCR, developed by the research group. The e-learning comprises the following modules:−Theoretical background and historical context of FICare and FCR;−Implementing FCR in practical settings;−Coaching of parents by healthcare professionals;−Explanation of the existing (co-)interventions SFR and family-centred care;−Understanding the principles of FCR and shared decision-making.−Training in FICare principles: the research group has designed training materials for effective collaboration and communication with parents, parent participation and shared decision-making. These training materials draw from both existing literature and the valuable insights available on the Canadian website dedicated to FICare (www.familyintegratedcare.com, accessed on 22 May 2022).

The training sessions consist of three modules, recorded as lectures, and are further discussed with the local staff by the respective FICare workgroups. The three modules covered are:−The theory and practical application of FICare and FCR;−The role of healthcare professionals within the context of FICare;−Emphasizing shared decision-making.

The intended audience for this training includes nurses, residents, and paediatricians, while also recommending paramedical staff to participate, at the very least, in the e-learning modules.Psychosocial support:−Facilitating contact and peer support for parents with veteran parents, both during admission and after discharge;−Support by paramedical staff (e.g., psychologist, preverbal speech therapist, physiotherapist, social worker).Environment of the neonatal ward:−FICare whiteboard/communication board, at which parents can track for example their participation progress and information on their infant;−Facilities for skin-to-skin contact and expressing human milk on the ward;−24/7 access to the ward;−Facilities to be (digitally) present during medical rounds (i.e., including video conference or telephone calls).

The NeoZorg application (Synappz Digital Healthcare^®^, Oss, The Netherlands)—originally developed for level 3 neonatal care and in use in the NICU in Amsterdam—was adapted and innovated especially for this study by our research group with extra information suitable for level 2 neonatal care. NeoZorg is a digital platform for parents with an infant in a neonatal ward, and can be used as an information and education medium for parents. The application provides a library with extensive expert information on relevant topics regarding neonatal care for parents. A special section is devoted to FICare and FCR videos. In addition to this library function, the application offers a diary function for parents to keep track of their child’s clinical condition and development during hospital stay. Parents can save notes and photographs and are provided space to write about their experiences and mental state. They can also enter clinical information such as growth, intake and human milk production, which can then be displayed in graphs. Through these functions, parents can keep up with the development of their infant’s clinical condition (in the Netherlands, real-time medical chart involvement is not usually provided). Alongside these functions, the application can send messages to parents with relevant information based on the stage of development and admission of their infant. Healthcare professionals can send parents pictures or short messages through a communal tablet that is present at the ward. As the application is purely intended to support parents, no study data will be gathered through the application. Given the high rates of smartphone ownership in the Netherlands, it is anticipated that every parent will have access to and be able to use the app. However, it is important to note that the app will not replace the paper diary in hospitals where it is already in use. Therefore, if parents choose not to utilise the app or face any other limitations, they can still opt for the traditional paper diary.

### 2.6. Control Treatment: Standard Neonatal Care without FCR

In the control period, standard neonatal care (SNC) will be provided. Although attention is given to parents in most current neonatal care, sometimes even based on the family-centred care principles, structural involvement of parents in care according to the FICare pillars is not yet implemented. Medical rounds are held between healthcare professionals without the (structural) presence of parents. Parents are updated by the nurses daily, or whenever a parent is present. Usually, parents are also updated at least weekly by their attending physician. Most care (both daily and medical care) for the infants is provided predominantly by the nurses. Parents usually have (unlimited) access to the ward but are not supported by the concept of FICare.

### 2.7. Interventional Treatment: Family-Centred Rounds Embedded in the FICare Principles

Families that are included during the intervention period will participate in family-centred rounds, while being supported by the principles of FICare as described in Section 2.5. In FCR, parents actively participate in the medical rounds with healthcare professionals and decisions are made based on shared decision-making, whenever appropriate. Not only are parents informed about the clinical condition of their child, they can ask questions and share their own valuable information on their child [20,21]. Giving the parents the opportunity to take on their irreplaceable role during medical rounds, and actively participate in the process of shared decision-making, requires more than merely an invitation: both parents and healthcare professionals need to receive the appropriate support and education. In the intervention period, this support is provided based on the four pillars of FICare. Also, parents will have access to the NeoZorg application (Synappz Digital Healthcare^®^).

### 2.8. Primary Outcome

The primary outcome of the study will be the level of parental stress at discharge. The Parental Stress Scale: Neonatal Intensive Care Unit (PSS:NICU) is the most widely used instrument to measure parental stress and stressors arising from hospitalisation of an infant on the NICU [31]. It measures the parental perception of stressors that are arising from the hospitalisation of their child, with a variety of health problems on any type of inpatient unit [32]. The design of this tool emphasises the 3 dimensions of the surroundings and experiences on an NICU: the environment of the unit (sights and sounds; 5 items), the appearance of the infant (14 items) and alterations in the parental role (7 items). The 3 subscales show an internal consistency (Cronbach’s α) of 0.73–0.94 [31,32]. A total of 26 items are scored from “not stressful at all” to “extremely stressful”, and both a subscore for each domain and a total score (ranging from 0 to 130) are given. Parents will be asked to fill out the PSS:NICU at discharge. The primary outcome will be defined as the mean difference in total PSS-NICU scores at discharge between the intervention and the control group.

### 2.9. Secondary Outcomes

Secondary outcomes of the study will be on the level of the infants, parents, healthcare professionals and organisation and on a possible underlying biological mechanism related to stress at parenteral level.

#### 2.9.1. Infant Outcomes

Infant outcomes include length of hospital stay, breastfeeding rates, growth, neurodevelopment at the (corrected) age of 12 months and saliva cortisol and buccal mucosal cells (see Section 2.9.5).

Neurodevelopment will be assessed using the Ages and Stages Questionnaire (ASQ, 3rd edition) [33]. With this tool, infant development can be assessed as reported by parents. The ASQ has been validated in a comparable population [34], and the validated Dutch version will be used for Dutch speaking participants [35]. The ASQ encompasses 5 domains: communicative, gross motor, fine motor, problem solving and adaptive skills. Each domain consists of 6 questions regarding the milestones that fit the age of the infant. Parents respond to the different items with “yes” (score = 10), “sometimes” (score = 5) or “not yet” (score = 0). For each domain, the score of the six items is summed, resulting in a domain score ranging from 0 to 60 and a total ASQ score with a maximum of 300 points. A positive screening is defined as scoring >2SD below the mean of the Dutch reference population in one domain, or scoring >1SD below the mean on more than one domain.

#### 2.9.2. Parental Outcomes

At discharge, parents’ experiences of the hospital admission will be evaluated in several domains. Their experience of shared decision-making will be measured by means of the Shared Decision Making Questionnaire (SDM-Q-9). This questionnaire contains 9 items, with a unidimensional structure, regarding patients’ experiences of the process of decision-making [36]. To adjust the SDM-Q-9 to a paediatric setting, “my doctor” will be replaced by “my baby’s doctor” and “my medical condition” will be changed to “my baby’s medical condition” with permission of the original authors. The SDM-Q-9 shows a good validity and reliability, with a Cronbach’s a of 0.94. Validation of the Dutch translation are in the same range, providing good reliability [37]. To be able to adjust for the preference parents may have in being involved in decision-making, the Control Preferences Scale (CPS) will be used. The CPS was originally created as a card-sorting exercise to assess the preferences an individual may have regarding the control in medical decision-making [38]. The CPS will be used in the form of a one-item questionnaire [39]. The item is answered on a 5-point Likert scale, and parents will be asked to fill out the CPS at discharge.

The CO-PARTNER tool will measure parental participation in the neonatal care and collaboration of parents with the medical team [40]. This tool can be used on the neonatal ward to describe the culture of a unit, measure the amount of FICare applied, define what parents can do in the care of their infant and in which care activities they collaborate with healthcare staff. It consists of 31 items within 6 domains: daily care (11 items), medical care (4 items), gathering information (3 items), advocacy (3 items), time spent with the infant (3 items) and closeness and comforting of the infant (7 items). Internal consistency of the domains varies from 0.558 to 0.938 and has a good convergent and divergent validity [40].

Parent–infant bonding will be measured using the Maternal Postnatal Attachment Scale (MPAS) [41]. The MPAS is a self-report tool containing 19 items with responses on a two- to five-point scale, reflecting the key experience of the mother-to-infant bond. The total score is formed by the sum of the 19 item responses. As each item score ranges from 1 (low attachment) to 5 (high attachment), to provide equal weighting of all items, the total score ranges from 19 to 95. The original version is based on a three-dimensional structure: pleasure in interaction with the infant, lack of negative feelings towards the infant, and sense of confidence and satisfaction in their competence as a parent [41,42,43]. Both the English and Dutch (translated) version show strong internal consistency [41,42]. For fathers, the Paternal Postnatal Attachment Scale (PPAS) will be used [44].

Parental mental wellbeing will be assessed longitudinally until the (corrected) age of 12 months of the infant. For this, three questionnaires will be used, sent to parents at admission (for parents of infants born <35 weeks), discharge and at the (corrected) ages of 3, 6 and 12 months of the infant. The subjects of mental wellbeing measured are stated below.

Post-traumatic stress symptoms will be assessed using the PCL-5. The PCL-5 is a diagnostic tool for post-traumatic stress disorders, adjusted to the DSM-5 [45,46]. The PCL-5 contains 20 items in a 4-factor model based on symptom clusters: intrusion, avoidance, negative alterations of cognitions and mood, and alterations in arousal and activity. The original (English) version shows a good internal consistency with a Cronbach’s a of 0.94, and has been validated in parents with satisfactory psychometric properties [47]. The PCL-5 has been translated to Dutch [48], and the translated tool shows an internal consistency similar to the original English version [49].

Depression and anxiety in parents will be measured using short forms from the PROMIS item bank. The PROMIS (Patient-Reported Outcomes Measurement Information System) item banks have been developed using both qualitative and quantitative approaches, aiming to develop more efficient PROMs that are valid, reliable and responsive [50]. The short forms contain fixed items chosen from an item bank of 6–121 items all measuring the same construct. The anxiety and depression short forms consist of 8 items each, and both show excellent internal consistency [51]. The scores of short forms are expressed in T-scores, which are standardised scores with a mean of 50 and a standard deviation of 10. The higher the (T) score, the more symptoms a parent experiences.

Outcomes at patient level (parents and infants) are listed in Table 1. Biological samples will be obtained as described under Section 2.9.5.

#### 2.9.3. Outcomes at Cluster Level

Outcomes for healthcare professionals include work engagement, autonomy and experiences in shared decision-making. The measurements of these outcomes are performed as follows.

As shared decision-making is a bidirectional process, in which both the patient and healthcare professional are involved, the healthcare professionals’ experience of the shared decision-making process will be measured. The SDM-Q-Doc, an adjusted version of the SDM-Q-9, will be used for this purpose. Similar to the version for patients (parents), the SDM-Q-Doc has 9 items in a unidimensional structure, and shows a good internal consistency with a Cronbach’s a of 0.88 [60]. The SDM-Q-Doc has been translated into Dutch [37]. To adjust the SDM-Q-Doc to our paediatric setting, “my patient” will be replaced by “the parent(s) of my patient”.

Although work pleasure is not equal to work engagement, engagement can be measured as an indicator for work pleasure. For this study, work engagement in healthcare professionals will be measured using the shortened version of the Utrecht Work Engagement Scale (UWES) [61]. The UWES-9 has 9 items (such as “At my work, I feel bursting with energy”) based on 3 subdomains (vitality, absorption and commitment). All items are scored on a 7-point frequency rating scale ranging from 0 (“never”) to 6 (“always”) [62]. The UWES-9 shows excellent internal consistency, with a Cronbach’s a of 0.93 [63].

By using the decision authority subscale of the Job Content Questionnaire (JCQ) [64], healthcare professionals’ autonomy will be assessed. The JCQ is a questionnaire with 24 items regarding aspects of work satisfaction, such as support by supervisors and work stability. The decision authority is a subscale that consists of 3 items (such as “My job allows me to make a lot of decisions on my own”), scored on a 4-point Likert scale (from “strongly disagree” to “strongly agree”) with which participants indicate at what level they agree with the statement. The total score in the subscale ranges from 3 to 12, with higher scores indicating more feelings of autonomy.

Organisational outcomes are duration and frequency of rounds, parental presence at rounds and cost-effectiveness. Outcome levels for healthcare professionals and organisations will be measured at start, halfway through and at the end of the study.

All outcomes at cluster level (healthcare professionals, cost-effectiveness) are listed in Table 2.

#### 2.9.4. Cost-Effectiveness

Cost-effectiveness will be analysed at the level of families (medical and productivity costs), organisations (work absence, length of infant’s hospital stay, hospital care costs) and healthcare professionals (productivity costs).

The social aspect of cost-effectiveness will be assessed by both the Productivity Cost Questionnaire and the Medical Cost Questionnaire of the institute for Medical Technology Assessment (iMTA PCQ and iMTA MCQ) [58,59]. The questionnaires will be slightly adjusted to fit the paediatric setting, and parents will be asked to fill out these questionnaires at the (corrected) age of 12 months of their infant. Healthcare professionals will be asked to fill out the iMTA PCQ at start, halfway through and at the end of the study.

#### 2.9.5. Biomarkers

To assess the effect of FICare on the stress hormone levels in human milk, samples of human milk will be collected at discharge and at the (corrected) age of 3 months of the infant. Mothers are requested to collect their milk from the first feeding moment in the morning at two timepoints. They are instructed to empty one breast in the morning before feeding their child. After mixing the milk, 2–10 mL will be donated in a sterile container that is provided and subsequently the collected human milk will be stored in the refrigerator at 2–8 °C. The composition (including macro- and micronutrients), immunological factors (ELISA analysis) and cortisol levels (LC-MS/MS) in human milk will be analysed.

To analyse the effect of FICare on the physiological stress response, cortisol levels in parents and infants will be assessed. Salivary samples will be collected at the (corrected) age of 3 months of the infant. On the measurement day, two samples of saliva (S1 and S2) are collected: one in the morning and one in the evening. The saliva is collected by chewing for 1 min on a swap (Salivette, Sarstedt, Rommelsdorf, Germany). S1 is obtained within 15 min after awakening in the morning, S2 is obtained before going to bed. By collecting two samples, we will make sure the cortisol day curve is captured. For infants, samples will be collected by parents before a feeding, by using a saliva swab designed for use in infants (Oracol Plus, Malvern Medical Developments, Worcester, United Kingdom). Salivary cortisol levels will be analysed by using isotope dilution liquid chromatography–tandem mass spectrometry (LC–MS/MS) [57,65].

Buccal mucosa samples of infants will be collected at discharge, and at the (corrected) age of 3 months by brushing a swab along the buccal mucosa [66]. Glucocorticoid receptor methylation rate will be analysed using the Mquant method [55].

The hair cortisol concentration (HCC) reflects long-term integrated cortisol levels, i.e., cortisol production over a prolonged period. As such, it is an index of chronic stress. Parents will be asked to collect hair samples by themselves at the (corrected) age of 3 months of the infant. Hair samples will be collected carefully with scissors as close as possible to the scalp at the posterior vertex of the head, since it has been found that this area of the scalp has the greatest growth cycle synchrony and exhibits the lowest intra-individual variability in HCC [67]. Approximately 100 hairs will be sampled. Cortisol and cortisone concentrations will be determined from the 1–3 cm segment of hair, representing hair growth over a 3-month period [68]. Hair samples will be analysed using a column switching LC–APCI–MS/MS assay [69].

### 2.10. Statistical Procedures

#### 2.10.1. Sample Size Calculation

Based on previous research, a mean difference of 13 points (0.5 standard deviations) on the total score of the PSS:NICU is expected between the intervention and control groups after implementation of FCR and FICare [25,26]. The stepped-wedge design will have a total of 7 steps of 3 months each. At each step, two hospitals will implement the FICare intervention. One step will be an implementation (or wash-out) period. With an expected amount of 25% missing data, a power of 80%, intra-cluster correlation of 0.01 and an eta of 0.5 times the estimated effect, we aim to include 600 infants and their parents divided over 10 clusters.

#### 2.10.2. Statistical Analysis

All analyses will be performed based on an intention-to-treat principle to minimise attrition bias. A sensitivity analysis with a per protocol analysis, including, for instance, only the infants and families who have not been transferred from the study site to another hospital before discharge home, will also be performed. Possible bias due to differential withdrawal or study drop-out will be assessed in this per protocol sensitivity analysis as well. We will consider performing a subgroup analysis based on clinical characteristics, to adjust for differences in (severity of) infants’ medical conditions.

Baseline characteristics of the intervention and control group will be compared using student *t*-tests for continuous normally distributed variables, and Chi-square tests for categorical variables. Non-parametric tests will be used for variables that do not follow a normal distribution. Missing data will be handled by multiple imputation chain equations (mice) [70] with adjustment for clustering [71]. Rubin’s rules will be applied to pool the results of the different imputed dataset [72]. As described and proposed by the Panel on Handling Missing Data in Clinical Trials, we will carry out sensitivity analyses with different missing data strategies adjusted for clustering [73]. We will consider missing completely at random, missing at random, and missing not at random approaches.

With a generalised linear mixed modelling (GLMM) analysis, we will analyse whether parental stress at discharge is decreased after the intervention. We will include the factor “cluster” (hospital) as random effects, and “step” will be included as fixed (time) effect. In a secondary analysis, adjustment can be applied for possible confounders, such as (medical) characteristics of the infant during hospital stay. Adjustment for possible time effects will be made.

## 3. Results

### 3.1. Trial Progress

Ethical approval by the Medical Ethics Review Committee (MEC-U, Nieuwegein, The Netherlands) was received on 6 December 2021. Recruitment of participants started on 7 March 2022, in all participating centres. The trial has been registered at www.clinicaltrials.gov (accessed on 31 Octobre 2022) under registration number NCT05343403. Trial progress and information can also be found on www.zorgevaluatienederland.nl/neopartner (accessed on 28 August 2023). Currently, all 10 centres have started enrolling patients. Recruitment is expected to be completed by December 2023.

### 3.2. Reporting and Publication

The ultimate goal is to advocate for the development of a national guideline on FICare implementation in neonatal units across the Netherlands, thereby enhancing the quality of care and support provided to both infants and their families. The research findings will be reported according to the CONSORT 2010 guidelines for stepped-wedge cluster-randomised controlled trials [28]. These publications will provide detailed insights into the results reaching a wide audience of healthcare professionals and policymakers. Our research team will present the study’s findings at national and international conferences related to neonatology, paediatrics, and family-centred or integrated care. By presenting the study findings at conferences and meetings, we can target professionals directly involved in neonatal care and influence their practice. We will collaborate with relevant professional associations in neonatal care and paediatrics to share the study results. These associations include the Dutch Society of Paediatrics, Dutch Section of Neonatology, and other related organisations. To raise awareness among the general public about the benefits of FICare and its potential impact on neonatal care, we will initiate a public awareness campaign through social media and collaborations with relevant health-focused organisations. By adopting these dissemination strategies, we aim to ensure that the study’s results reach the various stakeholders, including parents, healthcare professionals, and policymakers.

## 4. Discussion

Although previous research has highlighted the importance of parental participation in neonatal care [16,24,25,26,74,75,76], the broad implementation of a care model in neonatal wards to achieve such collaboration with parents still lags behind [12]. In a survey with 400 parents held by our research group prior to this study, the FICare principles were, as reported by parents, hardly implemented in the Netherlands [77]. Moreover, although parental participation in medical rounds can play a crucial role in empowering parents [16], evidence of the best practice and effects of family-centred rounds on neonatal wards is missing. In an era of staff shortage where time seems to be perpetually scarce, a readily available blueprint to ameliorate parental participation on neonatal wards could help spread programs such as family-integrated care more easily. The implementation plan that is described in this study is tailored to the Dutch setting and developed in close collaboration with the neonatal parent and patient organisation (Care4Neo), putting parental involvement into practice from the very start. The protocol outlined in this manuscript provides an overview of a novel trial design to implement and sustain FICare in neonatal wards, while investigating the (expected beneficial) effects. Alongside the outcomes measured at the individual level (infants and parents), the outcomes measured at cluster level (healthcare professionals, adherence to the intervention, cost-effectiveness) could provide meaningful information for management in deciding on policy strategies.

By using the stepped-wedge cluster-randomised trial (swCRT) design, this study aims to deliver robust evidence on the effects of FCR as part of the FICare concept. Apart from the practical and ethical advantages the swCRT design offers for this particular intervention, the methodological and/or statistical approach also enlarges generalizability. For example, the design makes accounting for possible (substantial) confounding effects of factors such as hospital culture, policies, architecture and size possible. Also, with the awareness of the importance of parental involvement possibly also growing naturally in neonatal care practices, the interaction of time is considered in the analysis.

Up till now, the effect of FICare on the physiological stress response in both infants and parents, using biomarkers of stress, has remained unexplored. It is known that psychological distress in lactating mothers during lactation can alter the composition of human milk regarding macro- and micronutrients, as well as the hormonal and immunological components [78,79,80]. Human milk is considered to improve infant health outcomes by facilitating the transmission of nutrients, hormones and cytokines from mother to child. The stress response is regulated to a great extent by glucocorticoids and previous research has shown that psychosocial interventions can affect cortisol levels in human milk [81]. However, the effect of interventions targeted on parental participation on such biomarkers of stress has not been described thus far. Therefore, the analysis of the cortisol levels in both human milk, hair and buccal saliva that are included in this trial might reveal answers and substantiate the beneficial effects of FICare on reducing stress.

Another novel evaluation includes a broad spectrum of the cost-effectiveness of FICare, beyond the evaluation of the infant’s length of stay [76]. Although reducing length of stay might evidently reduce healthcare costs, the effect of FICare on length of stay differs between studies [74,76,82]. The socio-economic effects of FICare, such as timing of resuming work by parents or work absence of healthcare professionals, have not been evaluated yet. By describing the cost-effectiveness, this study could guide hospital management to support implementation of FICare on their neonatal wards.

Changing hospital care culture and professional working methods, as occurs with implementing FICare, might impact healthcare professionals. Qualitative research evaluating the experiences of healthcare professionals with the FICare concept reveals that nurses indeed experience an alteration in their (professional) role, shifting from a professional caregiver to more of a supporting and educational role [83]. While nurses describe their new role as enhancing the parent–staff relationship, the process itself of changing might create challenges [17]. The effect of such interventions on the overall work pleasure and engagement in work of healthcare professionals remains unknown.

An obvious limitation of this study is the non-blinding of participants (both parents and healthcare professionals) and researchers. The design of the study and the nature of the intervention makes for an unfeasible setting for blinding. However, the effects of the non-blinding design are expected to be very low on the patient (infant and parent) level, as researchers cannot influence the medical facts or responses given by parents. There is a possible bias introduced by not blinding healthcare professionals (as some could have been supporters of the FICare concept beforehand, influencing their judgement and responses to questionnaires), but this is also expected to be of low relevance. As the other outcomes on cluster level are all objective (such as cost-effectiveness), bias is not expected on this level.

Another point worth mentioning is the possible differences in the (quality of) execution of the intervention between hospitals. Many principles of both the FCR and the FICare rely on human factors such as the communicative and collaborative skills of the healthcare professionals. In an ideal setting, in the absence of any financial or time restrictions, a more extensive training, such as that described in Finnish studies, would be incorporated in the implementation strategy [75]. However, as it is clear that steps need to be made based on current practices and the possible positive effects of parental participation [12,25,26], we chose to work with an implementation strategy that is feasible for most hospitals regardless of size or staff shortage. The results of this trial might implicate the need for further improvement or extension of this strategy, for example, creating a “FICare-plus” strategy for high income countries’ level 2 wards, comparable to the work described by a Spanish research group [84] for level 3 settings and the work published in Canada on Alberta FICare [76].

Lastly, in this study, only participants that are willing and able to fill in the questionnaires in English or Dutch are included. We conclude this as a significant limitation of the study, as a total of 149 languages or dialects are spoken at home in the Netherlands, with more than 8 percent of the Dutch population speaking a different language than Dutch at home [85]. Unfortunately, due to the limited availability of the outcome measurement tools in languages other than Dutch or English, we were not able to overcome this limitation at this point in time. However, the FICare implementation will be accessible to all parents whose infant is admitted to the neonatal wards of the participating centres. It is our opinion that the goal of FICare should be to empower every parent according to their preferences, needs and abilities. As such, there is not one single form of FICare that suits every parent, but individual tailoring outside of the healthcare professional’s perspective or habits could form a challenge. Further research should thus focus on gaining more knowledge on what parents of all different cultures, socio-economic statuses and personal situations need in order to participate in the manner they prefer and are able to.

## 5. Conclusions

To our knowledge, the neoPARTNER study is the first stepped-wedge cluster-randomised controlled trial in level two neonatal wards to assess the impact of FICare with FCR on (longitudinal) parental mental health, neonatal health (including biomarkers of stress) and organisational outcomes. This will enable all stakeholders in neonatal care to gather relevant and useful data to optimise and further humanise neonatal care.

## Figures and Tables

**Figure 1 children-10-01482-f001:**
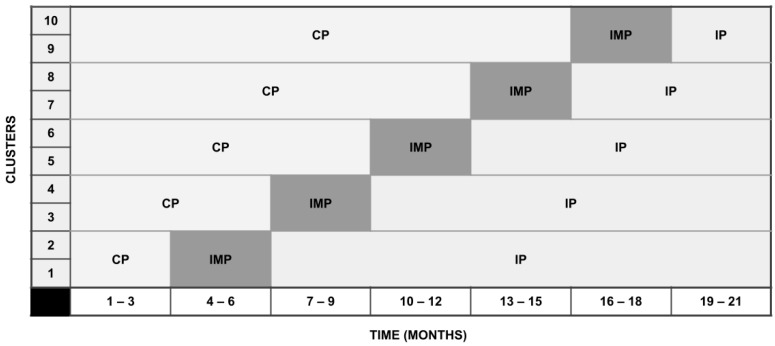
Visual overview of the study design. CP = control period; IMP = implementation period; IP = intervention period.

**Table 1 children-10-01482-t001:** Study procedures: parents and infants.

Timing	Subject	Outcome	Tool/Unit
Admission ^1^	Parents	Parental stress levels ^2^	PSS:NICU [31,32]
		Depression	PROMIS [50,51]
		Anxiety	PROMIS [50,51]
		PTSD	PCL-5 [45,46,48]
	Infant	Baseline characteristics	N/A
Discharge	Parents	Demographics	General questionnaire
		Parental stress levels	PSS:NICU [31,32]
		SDM	SDM-Q-9 [36,37]
		SDM preference	CPS [39]
		Depression	PROMIS [50,51]
		Anxiety	PROMIS [50,51]
		PTSD	PCL-5 [45,46,48]
	Mothers	Human milk biofactors	See text
	Infant	Breastfeeding rates	Index of breastfeeding [52,53]
		Length of stay	days
		Growth	Weight gain velocity [54]
		Glucocorticoid receptor methylation rate in buccal mucosal cells	Mquant method [55]
CA of 3 months of the infant	Parents	Follow-up characteristics	General questionnaire
		Depression	PROMIS [50,51]
		Anxiety	PROMIS [50,51]
		PTSD	PCL-5 [45,46,48]
		Hair cortisol	LC-MS/MS [56,57]
		Salivary cortisol	LC-MS/MS [56,57]
	Mothers	Human milk biofactors	See text
	Infant	Glucocorticoid receptor methylation rate in buccal mucosal cells	Mquant method [55]
		Salivary cortisol	LC-MS/MS [56,57]
CA of 6 months of the infant	Parents	Follow-up characteristics	General questionnaire
		Depression	PROMIS [50,51]
		Anxiety	PROMIS [50,51]
		PTSD	PCL-5 [45,46,48]
CA of 12 months of the infant	Infant	Neurodevelopment	ASQ [33,35]
		Medical costs	iMTA MCQ [58]
	Parents	Follow-up characteristics	General questionnaire
		Productivity costs	iMTA PCQ [59]
		Medical costs	iMTA MCQ [58]
		Depression	PROMIS [50,51]
		Anxiety	PROMIS [50,51]
		PTSD	PCL-5 [45,46,48]

SDM = shared decision-making; N/A = not applicable; CA = corrected age; PTSD = post-traumatic stress disorder. ^1^ Measurement at admission will only be performed in (parents of) infants born <35 weeks of gestation. ^2^ At admission, PSS:NICU questionnaire will only be sent to parents of infants who were born at an NICU.

**Table 2 children-10-01482-t002:** Study procedures: healthcare professionals and organisations.

Timing	Subject	Outcome	Tool/Unit
At start, halfway through and end of study	HCP	Demographics	General questionnaire
		Work engagement	UWES-9 [61]
		Autonomy	Subscale of JCQ [64]
		SDM	SDM-Q-Doc [37,60]
		Productivity costs	iMTA PCQ [59]
	Organisation	Work absence	Percentages of absenteeism
		Parental presence at rounds	No. of parents present, no. of rounds
		Duration of rounds	Minutes

HCP = healthcare professional; SDM = shared decision-making.

## Data Availability

The data presented in this study will be available on request from the corresponding author. The data are not publicly available due to privacy reasons.

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
