# Peer review of "Collaborating to Improve Neonatal Care: ParentAl Participation on the NEonatal Ward—Study Protocol of the neoPARTNER Study"

_children, 2023, doi:10.3390/children10091482_

Round 1
Reviewer 1 Report
Hoeben et al.'s study protocol addresses an important research question that will aid in the improvement of neonatal care delivery. The protocol is both scientifically and ethically sound. I have the following minor suggestions to improve the protocol's readability.
Because NICU-level classifications differ across countries, it would be helpful to describe what Level-II NICU in The Netherlands corresponds to, and how it compares to NICU classifications in the United States and Canada, for example.
Please define post-intensive care/high-care patients, high-care, and medium-care patients in the inclusion criteria. Also, please explain why you need written informed consent from both parents rather than just one.
While it is commendable that the study will collaborate with a Dutch neonatal parent and patient advocacy organization (Care4Neo), please describe the level of parental involvement in the development of this protocol in detail. It would be beneficial to understand the strategy for engaging with parents at each stage of the research process, including the dissemination of findings. Please also describe the process for selecting parent representatives for working groups.
The use of the NeoZorg application remains unclear, as does the reason for restricting access to this app to the intervention arm. Also, describe whether the app's content and quality have been formally evaluated thus far. Furthermore, given the abundance of neonatal apps available for free, explain why this particular app was chosen. Will additional data be collected to see if parents are using any other apps? Finally, without knowing the smartphone ownership rates in The Netherlands, I am curious about what happens if parents are unable to access this app.
Please provide more information about the learning modules for healthcare professionals, including the source, mode of instruction, and content quality.
Please provide the psychometric properties of all standardized instruments used in this study, with a focus on the local context.
Please clarify whether or not a detailed statistical plan will be published prior to analysis. If not, please include a detailed statistical plan as an appendix to this protocol.
Please provide more information about dissemination, especially to parents and other relevant stakeholders such as policymakers.
Author Response
REVIEWER #1
Response to the reviewer: Thank you for your kind words and thorough review of our manuscript. Below are our answers to your questions and/or remarks with reference to the adjusted sections in the manuscript where applicable.
Hoeben et al.'s study protocol addresses an important research question that will aid in the improvement of neonatal care delivery. The protocol is both scientifically and ethically sound. I have the following minor suggestions to improve the protocol's readability.
- Because NICU-level classifications differ across countries, it would be helpful to describe what Level-II NICU in The Netherlands corresponds to, and how it compares to NICU classifications in the United States and Canada, for example.
We decided, for simplicity reasons, to only specify the levels of neonatal intensive care according to the structure provided by the AAP. Consequently, only level 2 neonatal wards are participating in this study.
We changed the following in section setting 2.3 paragraph 4 (text-lines 164-170):
“In the Netherlands, the level of neonatal care is classified similarly to the structure provided by the American Academy of Pediatrics (AAP) [29]. Level 2 NICUs, also called special care nurseries, care for babies born at a gestational age of 32 weeks or more. These facilities are suited for babies who have moderate medical issues and are expected to recover fairly quickly. All the participating centres are level 2 neonatal wards. The average number of patients admitted to the neonatology department for more than 7 days per participating centre varies from 80 to 300 patients per year.”
- Please define post-intensive care/high-care patients, high-care, and medium-care patients in the inclusion criteria.
In addition to the additional text described above, in section 2.4. (Study population), paragraph 4, we have added the following (additional text lines 172-181):
“In 2021, in the Netherlands, approximately 15% of (live) born infants were born preterm and/or small for gestational age, accounting for 25.018 infants [30]. According to Dutch policies, infants born with a gestational age <32 weeks and/or birthweight <1200 grams or in need of intensive care (e.g. cardiorespiratory support) are born in/or treated at a NICU (level 3). In a Dutch level 2 neonatal ward, infants can receive non-invasive respiratory support, can have central venous catheters and can receive multiple medications. It is quite common for infants to be transferred between different hospitals prior to their final discharge to home.”
And in section 2.5. (Inclusion criteria), paragraph 5 (text-lines 183-184):
“Recruited patients will be patients admitted to a level 2 neonatal ward for 7 days or more.”
- Also, please explain why you need written informed consent from both parents rather than just one.
We added a sentence in section 2.4.1 (Inclusion criteria), text lines 189-190:
“Written informed consent of both parents/legal guardians (compliant to the regulations of the Central Committee on Research Involving Human Subject).”
- While it is commendable that the study will collaborate with a Dutch neonatal parent and patient advocacy organization (Care4Neo), please describe the level of parental involvement in the development of this protocol in detail. It would be beneficial to understand the strategy for engaging with parents at each stage of the research process, including the dissemination of findings.
We added a new section on the collaboration with Care4Neo in section 2.1 (Patient organisation involvement), paragraph 3, text lines 129-147:
“The Dutch patient organisation (Care4Neo) is actively involved in all stages of study, starting from the study design to the implementation of FICare in the participating hospitals, project evaluation and dissemination of the results. The Guideline for Reporting Involvement of Patients and the Public (GRIPP2 short form) is followed [27]. In the grant proposal, two representatives from Care4Neo (S.O-B., M.V.) were included in the project group, ensuring that the patient perspective and relevance were duly considered.
Regarding the study protocol and outcomes measured in the trial, a national survey was conducted together with Care4Neo to prioritise outcomes based on parents’ perspectives. The survey was disseminated through social media platforms of Care4Neo. Care4Neo was actively involved in the final selection of measurement methods and questionnaires. S.O.-B. and M.V., together with Care4Neo’s Parent Advisory Board consisting of experienced parents, thoroughly reviewed the questionnaires, including explanatory texts, and provided input on how to inform parents about any abnormal scores on the questionnaires during study participation. Additionally, Care4Neo played a key role in revising the patient information forms. During the trial, Care4Neo is involved in the implementation of FICare by coordinating the peer-to-peer support programme in each participating centre.”
- Please also describe the process for selecting parent representatives for working groups.
We added the following sentence in section 2.5 (Procedure), paragraph 5, text lines 223-233:
“To ensure feasibility and sustainability of the intervention in all participating sites, a FICare working group will be formed at each site. Each participating centre is responsible for the recruitment of participants of their own FICare working group. To achieve the broadest possible representation, FICare working groups will include neonatal nurses, paediatricians, veteran parents and other neonatal healthcare professionals (such as psychologists or physiotherapists). Veteran parent(s) will be recruited through the patient organisation (Care4Neo) or via the outpatient clinic by paediatricians. Training and guidance of the veteran parents will be coordinated by Care4Neo. The FICare working group will be responsible for tailoring the intervention to the specific neonatal ward, in cooperation with the research team, and the planning and execution of the staff and parent training.”
- The use of the NeoZorg application remains unclear, as does the reason for restricting access to this app to the intervention arm. Also, describe whether the app's content and quality have been formally evaluated thus far. Furthermore, given the abundance of neonatal apps available for free, explain why this particular app was chosen. Will additional data be collected to see if parents are using any other apps? Finally, without knowing the smartphone ownership rates in The Netherlands, I am curious about what happens if parents are unable to access this app.
The NeoZorg application is adapted and innovated by our study group to be used in our study in the intervention period. A special section contains information about Family Integrated Care and Family Centred Rounds with instruction videos and therefore, to prevent cross-contamination, the NeoZorg app is only offered to parents in participating hospitals in the intervention period.
We have added this information in section 2.5 (Procedure), paragraph 6, text lines 283-306:
“The NeoZorg application (Synappz Digital Healthcare®) – originally developed for level 3 neonatal care and in use in the NICU in Amsterdam – was adapted and innovated especially for this study by our research group with extra information suitable for level 2 neonatal care. NeoZorg is a digital platform for parents with an infant in a neonatal ward, and can be used as an information and education medium for parents. The application provides a library with extensive expert information on relevant topics regarding neonatal care for parents. A special section is devoted to FICare and FCR videos. In addition to this library function, the application offers a diary function for parents to keep track of their child's clinical condition and development during hospital stay. Parents can save notes and photographs and are provided space to write about their experiences and mental state. They can also enter clinical information such as growth, intake and human milk production, which can then be displayed in graphs. Through these functions, parents can keep up with the development of their infant’s clinical condition (as in the Netherlands, real-time medical chart involvement usually is not provided). Alongside, the application can send messages to parents with relevant information based on the stage of development and admission of their infant. Healthcare professionals can send parents pictures or short messages through a communal tablet that is present at the ward. As the application is purely intended to support parents, no study data will be gathered through the application. Given the high rates of smartphone ownership in the Netherlands, it is anticipated that every parent will have access to and be able to use the app. However, it is important to note that the app will not replace the paper diary in hospitals where it is already in use. Therefore, if parents choose not to utilise the app or face any other limitations, they can still opt for the traditional paper diary.”
- Please provide more information about the learning modules for healthcare professionals, including the source, mode of instruction, and content quality.
We have added and changed some sentences in section 2.5. (Procedure), paragraph 6, text lines 247-268:
- “Education of healthcare professionals:
- Comprehensive e-learning on FICare and FCR, developed by the research group. The e-learning comprises the following modules:
- Theoretical background and historical context of FICare and FCR;
- Implementing FCR in practical settings;
- Coaching of parents by healthcare professionals;
- Explanation of the existing (co-)interventions SFR and Family Centred Care;
- Understanding the principles of FCR and shared decision-making.
- Training on FICare principles: the research group has designed training materials for effective collaboration and communication with parents, parent participation and shared decision-making. These training materials draw from both existing literature and the valuable insights available on the Canadian website dedicated to FICare (www.familyintegratedcare.com).
The training sessions consist of three modules, recorded as lectures, and are further discussed with the local staff by the respective FICare workgroups. The three modules covered are:
- The theory and practical application of FICare and FCR;
- The role of healthcare professionals within the context of FICare;
- Emphasizing shared decision-making.
The intended audience for this training includes nurses, residents, and paediatricians, while also recommending paramedical staff to participate, at the very least, in the e-learning modules.”
- Please provide the psychometric properties of all standardized instruments used in this study, with a focus on the local context.
We have added the psychometric properties of all standardized instruments in section 2.9. (Outcomes), paragraph 8 through 12, text lines 354-575.
- Please clarify whether or not a detailed statistical plan will be published prior to analysis. If not, please include a detailed statistical plan as an appendix to this protocol.
We will publish a statistical analysis plan (SAP) prior to the end of the inclusion of our patients in the trial register (clinicaltrials.gov, registration number NCT05343403).
- Please provide more information about dissemination, especially to parents and other relevant stakeholders such as policymakers.
We extended section 3.2. (Reporting and publication), paragraph 13, text lines 551-567:
“The ultimate goal is to advocate for the development of a national guideline on FICare implementation in neonatal units across the Netherlands, thereby enhancing the quality of care and support provided to both infants and their families. The research findings will be reported according to the CONSORT 2010 guidelines for stepped wedge cluster randomised controlled trials [28]. These publications will provide detailed insights into results reaching a wide audience of healthcare professionals and policymakers. Our research team will present the study's findings at national and international conferences related to neonatology, paediatrics, and family-centred or integrated care. By presenting the study findings at conferences and meetings, we can target professionals directly involved in neonatal care and influence their practice. We will collaborate with relevant professional associations in neonatal care and paediatrics to share the study results. These associations include the Dutch Society of Paediatrics, Dutch Section of Neonatology, and other related organisations. To raise awareness among the general public about the benefits of FICare and its potential impact on neonatal care, we will initiate a public awareness campaign through social media and collaborations with relevant health-focused organisations. By adopting these dissemination strategies, we aim to ensure that the study's results reach various stakeholders, including parents, healthcare professionals, and policymakers.”

Reviewer 2 Report
The authors have proposed excellent study design for this very important issue of family centered rounds and family integrated care.
Authors may want to clarify certain aspects of the manuscript so it would be easier for readers to understand the study design.
1. Study design: Stepped wedge-controlled design is an excellent idea to avoid cross contamination but also loses individual randomization. Also it is important that all sites have similar patient population. Also, the groups may not be matching if one site is bigger than another site. Additionally, authors may want to write clinically in the design that all qualified babies for that site after washout period will be included in the interventional group and all babies prior to training will be included as control. Authors may also want to address how long is the training and what happens during the training period.
2. Authors may want to clarify what population of patients get admitted to NICU that will qualify. Are there any transfers from another level 3 units involved? How are authors going to distinguish between sick patients compared to only premature babies needing to stay in NICU for feeding or oxygen?
3. Study settings: Authors may want to mention how many average range of admissions are in level 2 units performing the study
4. Inclusion: Please describe or reference what do the authors mean by high care and medium care?
5. How long will it take to train a parent for FIC? Is 7 days good enough for assessing the results?
6. Authors state that all healthcare professionals are eligible to participate. IS there any plan to train all healthcare professionals?
7. Exclusion: What level of reading or education is necessary to fill out questionnaire? Why are authors excluding patients with other languages? What proportion of patients speak other languages? Especially immigrant population speaking other languages and illiterate parents are at much higher risk for having anxiety. Would this study not benefit them?
8. Would it be possible to describe who would be providing the actual training to parents?
9. Is the standard practice in in these units not to include parents in clinical rounds as majority of centers especially in US are moving towards family centered care models? Also do all parents not receive psychosocial support as standard practice? Do mothers not get evaluated for post partum depression screening?
10. Please describe about IRB approval, consenting and clinicaltrials.gov number in method section
Author Response
REVIEWER #2
Response to the reviewer: Thank you for reviewing our manuscript and providing us with your complimentary words and valuable suggestions.
The authors have proposed excellent study design for this very important issue of family centered rounds and family integrated care.
Authors may want to clarify certain aspects of the manuscript so it would be easier for readers to understand the study design.
- Study design: Stepped wedge-controlled design is an excellent idea to avoid cross contamination but also loses individual randomization. Also it is important that all sites have similar patient population. Also, the groups may not be matching if one site is bigger than another site. Additionally, authors may want to write clinically in the design that all qualified babies for that site after washout period will be included in the interventional group and all babies prior to training will be included as control.
We do agree with you that our stepped wedge study design is the best study design for conducting this research question. Due to the nature of the intervention, which involves changes to unit-level provision of care (medical rounds) and interaction between participants, there is a risk of cross-contamination on the ward Therefore, to avoid contamination of patients and staff, the stepped wedge cluster randomised controlled trial design was selected. The stepped wedge design takes into account changes in care and time (the power of a stepped wedge design decreases as the study duration increases because the influence of time becomes more significant). Moreover, as every site also functions as its’ own control group, we do not expect differences in sizes of sites to form a methodological problem.
Authors may also want to address how long is the training and what happens during
the training period.
In the three months preceding the intervention period (the implementation period), FCR and FICare principles will be implemented on the neonatal wards of the participating hospitals and all health care professionals will be trained with the FICare principles.
As is written in section 2.5. (Procedure), paragraph 5 (text-lines 219-233).
- Authors may want to clarify what population of patients get admitted to NICU that will qualify. Are there any transfers from another level 3 units involved?
We decided, for simplicity reasons, to only specify the levels of neonatal intensive care according to the structure provided by the AAP. Consequently, only level 2 neonatal wards are participating in this study.
We changed the following in section setting 2.3 paragraph 4 (text-lines 164-170):
“In the Netherlands, the level of neonatal care is classified similarly to the structure provided by the American Academy of Pediatrics (AAP) [29]. Level 2 NICUs, also called special care nurseries, care for babies born at a gestational age of 32 weeks or more. These facilities are suited for babies who have moderate medical issues and are expected to recover fairly quickly. All the participating centres are level 2 neonatal wards. The average number of patients admitted to the neonatology department for more than 7 days per participating centre varies from 80 to 300 patients per year.”
- How are authors going to distinguish between sick patients compared to only premature babies needing to stay in NICU for feeding or oxygen?
We added the following in section 2.10.2. (Statistical analysis), paragraph 13, text-line 525-526:
“We will consider performing a subgroup analysis based on clinical characteristics, to adjust for differences in (severity of) infants’ medical conditions.”
- Study settings: Authors may want to mention how many average range of admissions are in level 2 units performing the study
We added information regarding average range of admissions in level 2 hospitals in section 2.3 (Setting), paragraph 4 (text-line 168-170):
“The average number of patients admitted to the neonatology department for more than 7 days per participating center varies from 80 to 300 patients per year.”
- Inclusion: Please describe or reference what do the authors mean by high care and medium care?
As described above at comment number 2, we have added information regarding Dutch levels of neonatal care in section 2.3. (Setting), paragraph 4.
- How long will it take to train a parent for FIC? Is 7 days good enough for assessing the results?
In the three months preceding the intervention period (the implementation period), FCR and FICare principles will be implemented on the neonatal wards of the participating hospitals. The FICare care is offered to parents, and they are coached and supported in their active parental role. It will not be a mandatory training that parents must follow. We aim with our implementation study of FICare to change the culture of the neonatology ward and, in this way, parents are perceived as part of the team.
- Authors state that all healthcare professionals are eligible to participate. Is there any plan to train all healthcare professionals?
We have provided more explanation regarding the training of healthcare professionals in section 2.5. (Procedures), paragraph 6, text-lines 248-254:
“- Comprehensive e-learning on FICare and FCR, developed by the research group. The e-learning comprises the following modules:
- Theoretical background and historical context of FICare and FCR;
- Implementing FCR in practical settings;
- Coaching of parents by healthcare professionals;
- Explanation of the existing (co-)interventions SFR and Family Centred Care;
- Understanding the principles of FCR and shared decision-making.
- Training on FICare principles: the research group has designed training materials for effective collaboration and communication with parents, parent participation and shared decision-making. These training materials draw from both existing literature and the valuable insights available on the Canadian website dedicated to FICare (www.familyintegratedcare.com).
The training sessions consist of three modules, recorded as lectures, and are further discussed with the local staff by the respective FICare workgroups. The three modules covered are:
- The theory and practical application of FICare and FCR;
- The role of healthcare professionals within the context of FICare;
- Emphasizing shared decision-making.
The intended audience for this training includes nurses, residents, and paediatricians, while also recommending paramedical staff to participate, at the very least, in the e-learning modules.”
- Exclusion: What level of reading or education is necessary to fill out questionnaire? Why are authors excluding patients with other languages? What proportion of patients speak other languages? Especially immigrant population speaking other languages and illiterate parents are at much higher risk for having anxiety. Would this study not benefit them?
Thank you for this insightful comment on a subject that we also believe is of great importance. In this study only participants that are willing and able to fill in the questionnaires in English or Dutch are included. However, although they can not participate in our study, the FICare implementation will be accessible to all parents whose infant is admitted to the neonatal wards of the participating centers despite their language.
To elaborate more clearly on this subject, we have added the following statements in the Discussion, paragraph 15 (text-lines 646-659):
“Lastly, in this study only participants that are willing and able to fill in the questionnaires in English or Dutch are included. We conclude this as a significant limitation of the study, as a total of 149 languages or dialects are spoken at home in the Netherlands, with more than 8 percent of the Dutch population speaking a different language than Dutch at home [85]. Unfortunately, due to the limited availability of the outcome measurement tools in other languages than Dutch or English, we were not able to overcome this limitation at this point in time. However, the FICare implementation will be accessible to all parents whose infant is admitted to the neonatal wards of the participating centres. It is our opinion that the goal of FICare should be to empower every parent according to their preferences, needs and abilities. As such, there is not one single form of FICare that suits every parent, but individual tailoring outside of the healthcare professional’s perspective or habits could form a challenge. Further research should thus focus on gaining more knowledge on what parents of all different cultures, socio-economic statuses and personal situations need to participate in the manner they prefer and are able to.”
Currently, our study group is also involved in the RISEinFAMILY project, in which FICare is implemented in different countries (such as Romania, Turkey, United Kingdom and Zambia) aiming to develop an implementation plan of FICare suited to different cultural and socio-economical settings.
- Would it be possible to describe who would be providing the actual training to parents?
During the study, the patient organisation Care4Neo is involved in the implementation of FICare by coordinating the peer-to-peer support programme in each participating center. There is not a special training for parents, we see it more as a culture change in the neonatal ward; parents are seen as team members and active participants Parents are supported and coached by health care professionals. They will not have to perform an exam in FICare or what so ever.
We added some information about the FICare education to parents in section 2.5 Procedure paragraph 6 (text-lines 243-245):
“Educational and thematic meetings (physical and/or digital) with other parents, led by either healthcare professionals (nurses, paediatricians and/or paramedic staff) or veteran parents.”
- Is the standard practice in in these units not to include parents in clinical rounds as majority of centers especially in US are moving towards family centered care models? Also do all parents not receive psychosocial support as standard practice?
Unfortunately up till now parents are not standard invited in the clinical rounds in neonatal wards in the Netherlands. From our previous survey amongst parents of preterm infants, parents reported that only half was invited to attend medical rounds, of whom only a third felt welcome to participate in shared decision-making.
Do mothers not get evaluated for post partum depression screening?
Although most pregnant women get screened on mental wellbeing prenatally, psychological evaluation is not standard practice in the postpartum period.
- Please describe about IRB approval, consenting and clinicaltrials.gov number in method section
This is mentioned in the result section (3.1. Trial progress), paragraph 13, text-lines 544-549:
“Ethical approval by the Medical Ethics Review Committee (MEC-U, Nieuwegein) was received on December 6th, 2021. Recruitment of participants started on March 7th, 2022, in all participating centres. The trial has been registered at www.clinicaltrials.gov under registration number NCT05343403. Trial progress and information can also be found on www.zorgevaluatienederland.nl/neopartner. Presently, all 10 centres have started enrolling patients. Recruitment is expected to be completed by December 2023.”
